# Avoiding bias when inferring race using name-based approaches

**Diego Kozlowski**[1]*, **Dakota S. Murray**[2], **Alexis Bell**[3], **Will Hulsey**[3], **Vincent Larivière**[4], **Thema Monroe-White**[3], **Cassidy R. Sugimoto**[5]

**1** DRIVEN DTU, Faculté des Sciences, de la Technologie et de la Médecine, University of Luxembourg, Esch-sur-Alzette, Luxembourg, **2** School of Informatics, Computing, and Engineering, Indiana University Bloomington, Bloomington, Indiana, United States of America, **3** Campbell School of Business, Berry College, Mt Berry, Georgia, United States of America, **4** École de bibliothéconomie et des sciences de l'information, Université de Montréal, Montréal, Québec, Canada, **5** School of Public Policy, Georgia Institute of Technology, Atlanta, Georgia, United States of America

* diego.kozlowski@uni.lu

**Data Availability Statement:** The data used for this article are available at https://sciencebias.uni.lu/app/ and https://github.com/DiegoKoz/intersectional_inequalities.

## Abstract

Racial disparity in academia is a widely acknowledged problem. The quantitative understanding of racial-based systemic inequalities is an important step towards a more equitable research system. However, because of the lack of robust information on authors' race, few large-scale analyses have been performed on this topic. Algorithmic approaches offer one solution, using known information about authors, such as their names, to infer their perceived race. As with any other algorithm, the process of racial inference can generate biases if it is not carefully considered. The goal of this article is to assess the extent to which algorithmic bias is introduced using different approaches for name-based racial inference. We use information from the U.S. Census and mortgage applications to infer the race of U.S. affiliated authors in the Web of Science. We estimate the effects of using given and family names, thresholds or continuous distributions, and imputation. Our results demonstrate that the validity of name-based inference varies by race/ethnicity and that threshold approaches underestimate Black authors and overestimate White authors. We conclude with recommendations to avoid potential biases. This article lays the foundation for more systematic and less-biased investigations into racial disparities in science.

## Introduction

The use of racial categories in the quantitative study of science dates from so long ago that it intertwines with the controversial origins of statistical analysis itself [1,2]. However, while Galton and the eugenics movement reinforced the racial stratification of society, racial categories have also been used to acknowledge and mitigate racial discrimination. As Zuberi [3] explains: "The racialization of data is an artifact of both the struggles to preserve and to destroy racial stratification." This places the use of race as a statistical category in a precarious position, one that both reinforces the social processes that segregate and disempower parts of the population, while simultaneously providing an empirical basis for understanding and mitigating inequities.

**Funding:** VL acknowledges funding from the Canada Research Chairs program, https://www.chairs-chaires.gc.ca/, (grant # 950-231768), DK acknowledges funding from the Luxembourg National Research Fund, https://www.fnr.lu/, under the PRIDE program (PRIDE17/12252781). The funders had no role in study design, data collection and analysis, decision to publish, or preparation of the manuscript.

**Competing interests:** The authors have declared that no competing interests exist.

Science is not immune from these inequities [4–7]. Early research on racial disparities in scientific publishing relied primarily on self-reported data in surveys [8], geocoding [9], and directories [10]. However, there is an increasing use of large-scale inference of race based on names [11], similar to the approaches used for gender-disambiguation [12]. Algorithms, however, are known to encode human biases [13,14]: there is no such thing as *algorithmic neutrality*. The automatic inference of authors' race based on their features in bibliographic databases is itself an algorithmic process that needs to be scrutinized, as it could implicitly encode bias, with major impact in the over and under representation of racial groups.

In this study, we use the self-declared race/ethnicity from the 2010 U.S. Census and mortgage applications as the basis for inferring race from author names on scientific publications indexed in the Web of Science database. Bibliometric databases do not include self-declared race by authors, as they are based on the information provided in publications, such as given and family names. Given that the U.S. Census provides the proportion of self-declared race by family name, this information can be used to infer U.S. authors' race given their family names. Name-based racial inference has been used in several articles. Many studies assigned a single category given the family or given name [15–19]. Other studies used the aggregated probabilities related with a name, instead of using a single label [20]. In this research, we assess the incurred biases when using a single label, i.e. thresholding. The main goal of this research is to define the most unbiased algorithm to predict a racial category given a name. We present several different approaches for inferring race and examine the bias generated in each case. The goal of the research is to provide an empirical critique of name-based race inference and recommendations for approaches that minimize bias. Even if prefect inference is not achievable, the conclusions that arise from this study will allow researchers to conduct more careful analyses on racial and ethnic disparities in science. Although the categories analysed are only valid in the U.S. context, the general recommendation can be extended to any other country in which the Census (or similar data collection mechanism) includes self-reported race.

## Racial categories in the U.S. Census

The U.S. Census is a rich and long-running dataset, but also deeply flawed and criticized. Currently it is a decennial counting of all U.S. residents, both citizens or non-citizens, in which several characteristics of the population are gathered, including self-declared race/ethnicity. The classification of race in the U.S. Census is value-laden with the agendas and priorities of its creators, namely 18th century White men who Wilkerson [21] refers to as "the dominant caste." The first U.S. Census was conducted in 1790 and founded on the principles of racial stratification and White superiority. Categories included: "Free White males of 16 years and upward," "Free White males under 16 years;" "Free White females," "All other free persons," and "Slaves" [22]. At that time, each member of a household was classified into one of these five categories based on the observation of the census-taker, such that an individual of "mixed white and other parentage" was classified into "All other free persons" in order to preserve the "Free White. . ." privileged status. To date, anyone classifying themselves as other than "non-Hispanic White" is considered a "minority." The shared ground across the centuries of census survey design and classification strata reflects the sustained prioritization of the White male caste [3,23].

Today, self-identification is used to assign individuals to their respective race/ethnicity classifications [24], per the U.S. Office of Management and Budget (OMB) guidelines. However, the concept of race and/or ethnicity remains poorly understood. For example, in 2000 the category "Some other race" was the third largest racial group, consisting primarily of individuals who in 2010 identified as Hispanic or Latino (which according to the 2010 census definition

refers to a person of Cuban, Mexican, Puerto Rican, South or Central American, or other Spanish culture or origin regardless of race). Instructions and questions which facilitated the distinction between race and ethnicity began with the 2010 census which stated that "[f]or this census, Hispanic origins are not races" and to-date, in the U.S. federal statistical system, Hispanic origin is considered to be a separate concept from race. However, this did not preclude individuals from self-identifying their race as "Latino," "Mexican," "Puerto Rican," "Salvadoran," or other national origins or ethnicities [25]. Furthermore, 6.1% of the U.S. population changed their self-identification of both race and ethnicity between the 2000 and 2010 censuses [26], demonstrating the dynamicity of the classification. The inclusion of certain categories has also been the focus of considerable political debate. For example, the inclusion of citizenship generated significant debates in the preparation of the 2020 Census, as it may have generated a larger nonresponse rate from the Hispanic community [27]. For this article, we attempt to represent the fullest extent of potential U.S.-affiliated authors; thereby, we consider both citizens and non-citizen.

The social function of the concept of race (i.e., the building of racialized groups) underpins its definition more than any physical traits of the population. For example, "Hispanic" as a category arises from this conceptualization, even though in the 2010 U.S. Census the question about Hispanic origin is different from the one on self-perceived race. While Hispanic origin does not relate to any physical attribute, it is still considered a socially racialised group, and this is also how the aggregated data is presented by the Census Bureau. Therefore, in this paper, we will utilize the term race to refer to these social constructions, acknowledging the complex relation between conceptions of race and ethnicity. But even more important, this conceptualization of race also determines what can be done with the results of the proposed models. Given that race is a social construct, inferred racial categories should only be used in the study of group-level social dynamics underlying these categories, and not as individual-level traits. Census classifications are founded upon the social construction of race and reality of racism in the U.S., which serves as "a multi-level and multi-dimensional system of dominant group oppression that scapegoats the race and/or ethnicity of one or more subordinate groups" [28]. Self-identification of racial categories continue to reflect broader definitional challenges, along with issues of interpretation, and above all the amorphous power dynamics surrounding race, politics, and science in the U.S. In this study, we are keenly aware of these challenges, and our operationalization of race categories are shaped in part by these tensions.

## Data

This project uses several data sources to test the different approaches for race inference based on the author's name. First, to test the interaction between given and family names distributions, we simulate a dataset that covers most of the possible combinations. Using a Dirichlet process [29], we randomly generate 500 multinomial distributions that simulate those from given names, and another 500 random multinomial distributions that simulate those from family names. After this, we build a grid of all the possible combinations of given and family names random distributions (250,000 combinations). This randomly generated data will only be used to determine the best combination of the probability distributions of given and family names for inferring race.

In addition to the simulation, we use two datasets with real given and family names and an assigned probability for each racial group. The data from the given names is from Tzioumis [30], who builds a list of 4,250 given names based on mortgage applications, with self-reported race. Family name data is based on the 2010 U.S. Census [31], which includes all family names with more than 100 appearances in the census, with a total of 162,253 surnames that covers

more than 90% of the population. For confidentiality, this list removes counts for those racial categories with fewer than five cases, as it would be possible to exactly identify individuals and their self-reported race. In those cases, we replace with zero and renormalize. As explained previously, changes were introduced in the 2010 U.S. Census racial categories. Questions now include both racial and ethnic origin, placing "Hispanic" outside the racial categories. Even if now "Hispanic" is not considered a racial category, but an ethnic origin that can occur in combination with other racial categories (e.g., Black, White or Asian Hispanic), the information about names and racial groups merge both questions into a single categorization. Therefore, the racial categories used in this research includes "Hispanic" as a category, and all other racial categories excluding people with Hispanic origin. The category "White" becomes "Non-Hispanic White Alone", and "Black or African American" becomes "Non-Hispanic Black or African American Alone", and so on. The final categories used in both datasets are:

- Non-Hispanic White Alone (*White*)

- Non-Hispanic Black or African American Alone (*Black*)

- Non-Hispanic Asian and Native Hawaiian and Other Pacific Islander Alone (*Asian*)

- Non-Hispanic American Indian and Alaska Native Alone (*AIAN*)

- Non-Hispanic Two or More Races (*Two or more*)

- Hispanic or Latino origin (*Hispanic*)

We test these data on the Web of Science (WoS) to study how name-based racial inference performs on the population of U.S. first authors. WoS did not regularly provide first names in articles before 2008, nor did it provide links between authors and their institutional addresses; therefore, the data includes all articles published between 2008 and 2019. Given that links between authors and institutions are sometimes missing or incorrect, we restricted the analysis to first authors to ensure that our analysis solely focused on U.S. authors. This results in 5,431,451 articles, 1,609,107 distinct U.S. first authors in WoS, 152,835 distinct given names and 288,663 distinct family names for first authors. Given that in this database, 'AIAN' and 'Two or more' account for only 0.69% and 1.76% of authors respectively, we remove these and renormalize the distribution with the remaining categories. Therefore, in what follows we will refer exclusively to categories *Asian*, *Black*, *Hispanic*, and *White*.

## Methods

### Manual validation

The data is presented as a series of distributions of names across race (Table 1). In name-based inference methods, it is not uncommon to use a threshold to create a categorical distinction: e.g., using a 90% threshold, one would assume that all instances of Juan as first name should be categorized as Hispanic and all instances of Washington as a given name should be categorized as Black. In such a situation, any name not reaching this threshold would be excluded (e.g., those with the last name of "Lee" would be removed from the analysis). This approach, however, assumes that the distinctiveness of names across races does not significantly differ.

To test this, we began our analysis by manually validating name-based inference at three threshold ranges: 70–79%, 80–89%, and 90–100%. We sampled 300 authors from the WoS database, 25 randomly sampled for every combination of racial category and inference threshold. Two coders manually queried a search engine for the name and affiliation of each author and attempted to infer a perceived racial category through visual inspection of their

**Table 1. Sample of family names (U.S. Census) and given names (mortgage data).**

| Type | Name | Asian | Black | Hispanic | White | Count |
|------|------|-------|-------|----------|-------|-------|
| Given | Juan | 1.5% | 0.5% | 93.4% | 4.5% | 4,019 |
| | Doris | 3.4% | 13.5% | 6.3% | 76.7% | 1,332 |
| | Andy | 38.8% | 1.6% | 6.4% | 53.2% | 555 |
| Family | Rodriguez | 0.6% | 0.5% | 94.1% | 4.8% | 1,094,924 |
| | Lee | 43.8% | 16.9% | 2.0% | 37.3% | 693,023 |
| | Washington | 0.3% | 91.6% | 2.7% | 5.4% | 177,386 |

professional photos and information listed on their websites and CVs (e.g., affiliation with racialized organizations such as *Omega Psi Phi Fraternity*, *Inc.*, *SACNAS*, etc.).

Fig 1 shows the number of valid and invalid inferences, as well as those for whom a category could not be manually identified, and those for whom no information was found. Name-based inference of Asian authors was found to be highly valid at every considered threshold. The inference of Black authors, in contrast, produced many invalid or uncertain classifications at the 70–80% threshold, but had higher validity at the 90% threshold. Similarly, inferring Hispanic authors was only accurate after the 80% threshold. Inference of White authors was highly valid at all thresholds but improved above 90%. This suggests that a simple threshold-based approach does not perform equally well across all racial categories. We thereby consider an alternative weighting-based scheme that does not provide an exclusive categorization but uses the full information of the distribution.

## Weighting scheme

We assess three strategies for inferring race from an author's name using a combination of their given and family name distributions across racial categories (Table 1). The first two aim at building a new distribution as a weighted average from both the given and family name racial distributions, and the third uses both distributions sequentially. In this section we

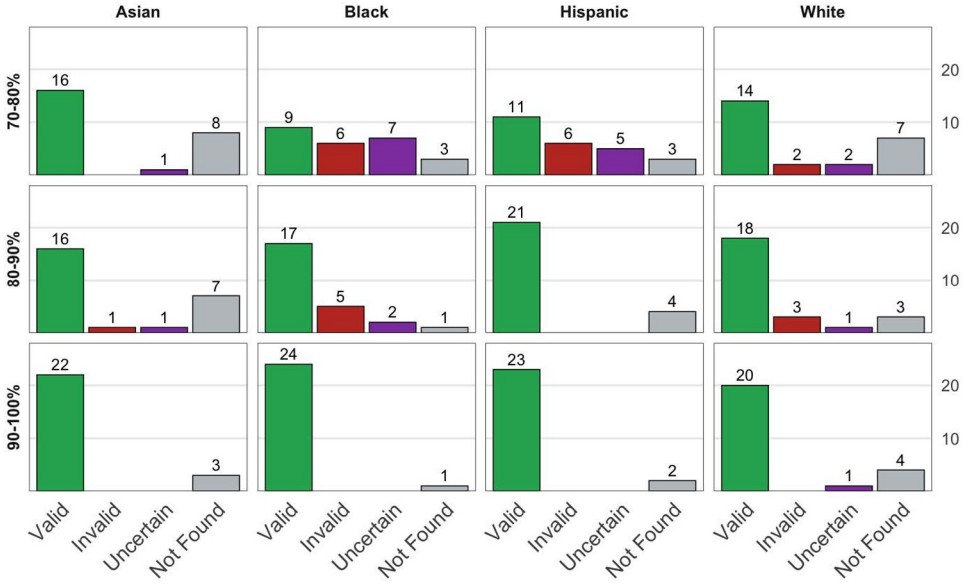

**Fig 1. Manual validation of racial categories.**

explain these three approaches and compare them to alternatives that use only given or only family name racial distributions.

The weighting scheme should account for the intuition that if the given (family) name is highly informative while the family (given) name is not, the resulting average distribution should prioritize the information on the given (family) name distribution. For example, 94% of people with Rodriguez as a family name identify themselves as Hispanic, whereas 39% of the people with the given name Andy identify as Asian, and 53% as White (see Table 1). For an author called Andy Rodriguez, we would like to build a distribution that encodes the informativeness of their family name, Rodriguez, rather than the relatively uninformative given name, Andy. The first weighting scheme proposed is based on the standard deviation of the distribution:

$$SD = \sqrt{\frac{1}{n-1}\sum_{i=1}^{n}(x_i - \bar{x})^2}$$

Where $x_i$ is in this case the probability associated with category $i$, and $n$ is the total number of categories. With four racial categories, the standard deviation moves between 0, for perfect uniformity, and 0.5 when one category has a probability of 1. The second weighting scheme is based on entropy, a measure that is designed to capture the informativeness of a distribution:

$$Entropy = -\sum_{i=1}^{n}P(x_i)logP(x_i)$$

Using these, we propose the following weight for both given and family names:

$$x_{weight} = \frac{f(x)^{exp}}{f(x)^{exp} + f(y)^{exp}}$$

with $x$ and $y$ as the given (family) and family (given) names respectively, $f$ is the weighting function (standard deviation or entropy), and $exp$ is the exponent applied to the function and a tuneable parameter. For the standard deviation, using the square function means we use the variance of the distribution. In general, the higher the $exp$ is set, the more skewed the weighting is towards the most informative name distribution. In the extreme, it would be possible to use an indicator function to simply choose the most skewed of the two distributions, but this approach would not use the information from both distributions. For this reason, we decided to experiment with $exp \in \{1,2\}$, which imply a trade-off between selecting the most informative of the two distributions, and using all available information.

Fig 2 shows the weighting of the simulated given and family names based on their informativeness, and for different values of the exponent. The horizontal and vertical axes show the highest value on the given and family name distribution, respectively. This means that a higher value on any axis corresponds with a more informative given/family name. The color shows how much weight is given to given names. When the exponent is set to two, both the entropy and standard deviation-based models skew towards the most informative feature, a desirable property. Compared to other models, the variance gives the most extreme values to cases where only one name is informative, whereas the entropy-based model is the most uniform.

## Information retrieval

The above weighting schemes result in a single probability distribution of an author belonging to each of the racial categories, from which a race can be inferred. One strategy for inferring race from this distribution is to select the racial category above a certain threshold, if any. A

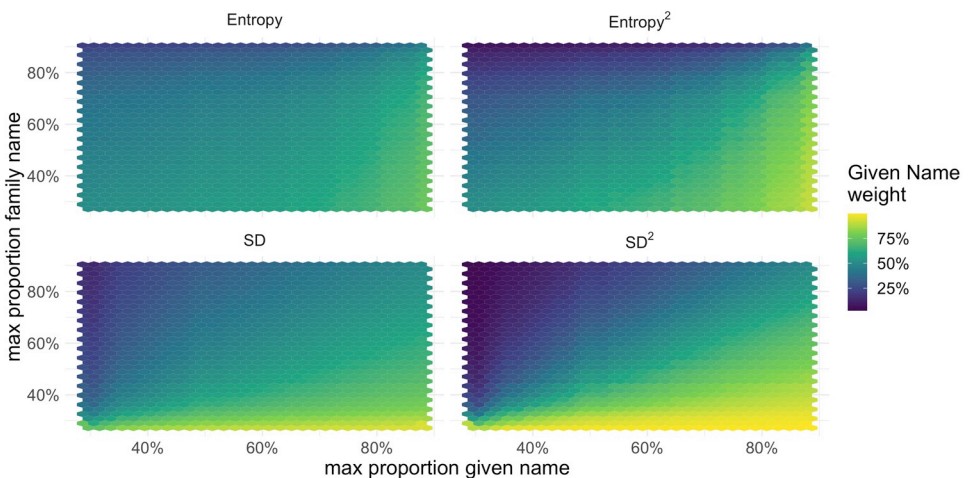

**Fig 2. Given names weight distribution by given and family name skewness.** Simulated data.

second strategy is to use the full distribution to weight the author across different racial categories, rather than assigning any specific category. We also consider a third strategy, which sequentially uses family and then given names to infer race.

We first retrieve all authors who have a family name with a probability of belonging to a specific racial group greater than a given threshold. This retrieves $N$ authors. Second, we retrieve the same number of authors as in the first step, $N$, using their given names. Finally, we merge the authors from both steps, removing duplicates who had both given and family names above the set threshold. This process results in between $N$ and $2N$ authors. There are several natural variations on this two-step method. For example, a percentage threshold could be used for both steps, or the first step could use given names, rather than family. We select family names first, because they are sourced from the larger and more comprehensive census data.

In summary, the following methods will be used on the empirical analysis.

1. Only family names with thresholding,

2. Only given names with thresholding,

3. Weighted average of given and family names using the variance as weighting scheme,

4. Two-step retrieval,

5. Fractional counting, for comparison.

## Results

### The effect of underlying skewness

Before comparing the results of the proposed strategies for using both given and family names, we present characteristics of these two distributions on the real data, and in relation to the WoS dataset. Table 2 shows the population distribution on the family names, based on the U. S. Census, and on the given names, based on the mortgage applications. Considering the U.S. Census data as ground truth, we see that the mortgage data highly over-represents the White population, particularly over-represents Asians, and under-represents Black and Hispanic populations; this likely stems from the structural factors (i.e., economic inequality, redlining, etc.) that prevent marginalized groups from applying for mortgages in the U.S. People may

**Table 2. Racial representation of family names (U.S. Census) and given names (mortgage data).**

| Racial group | Family names | Given names |
|---|---|---|
| Asian | 5.0% | 6.3% |
| Black | 12.4% | 4.2% |
| Hispanic | 16.5% | 6.9% |
| White | 66.1% | 82.6% |

also choose to self-report a different racial category when responding anonymously to the census bureau than when applying for a mortgage loan. Due to this bias in the distribution of given names, we decided to implement a normalized version of the given names racial distribution. This was obtained by computing the total number of cases for each racial group in each dataset, and the expansion factor for each group, obtained by the ratio between the total number of cases in the census data (family names) with respect of the Mortgage data (given names). We use this expansion factor to multiply the cases of each group for each name, and finally divide by the total number of cases in each name to have the proportion of each racial group on each name. By doing this, the average distribution of the given names data matches the one in the U.S. Census. In what follows, we use both the normalized and not normalized version of given names, for comparison.

Both given and family names share a characteristic not considered in our simulated data: the informativeness of names varies across racial groups. Inferring racial categories based on a set threshold will, then, produce biased results as typical names of one racial category are more informative, and thus more easily meet the threshold, than another. Fig 3 shows the ratio of the proportion of each racial group for different thresholds with respect to a 0% threshold, which implies fractional counting and the closest we can get to ground truths with the available information. This figure shows how the representation of inferred races changes based on the assignment threshold used. Increasing the threshold results in fewer total individuals returned (top), as some names are not sufficiently informative. For family names, only a small proportion of the population remains at the 90% threshold. The Asian population is highly over-represented between the 90% and 96% threshold, after which they suddenly become underrepresented. The White population is systematically over-represented for any threshold, whereas the Black population is systematically under-represented. The Hispanic population is over-represented between the 65% and 92% threshold and under-represented after. Similar results are observed based on given names. Again, the Asian population is highly over-represented after the 96% threshold, whereas the White population is over-represented across nearly all thresholds and the Black and Hispanic population were under-represented across all thresholds. With given names, the White population is systematically overestimated for every threshold until 96%, where the Asian population is suddenly overestimated to a high degree. The fact that Asian, and to some degree Hispanic, populations have more informative given and family names reflects their high degree of differentiation from other racial groups in the U.S.; White and Black populations in the United States, in contrast, tend to have more similar names (as verified in [32]). Given that the White population is larger than the Black population in the U.S., the use of a threshold (and assigning all people with that name to a single category), generates a Type I error on Black authors, and Type II error on White authors, thereby overestimating the proportion of White authors. Likewise, the descendants of African chattel slavery in the U.S. were assigned names by their rapists/slavers as a form of physical bondage and psychological control. Furthermore, family members who had been sold away, often retained their names, including those of U.S. Presidents George Washington and James Monroe, in hopes of making it easier to reunite with loved ones. [33–35]. After the 1960's however, and

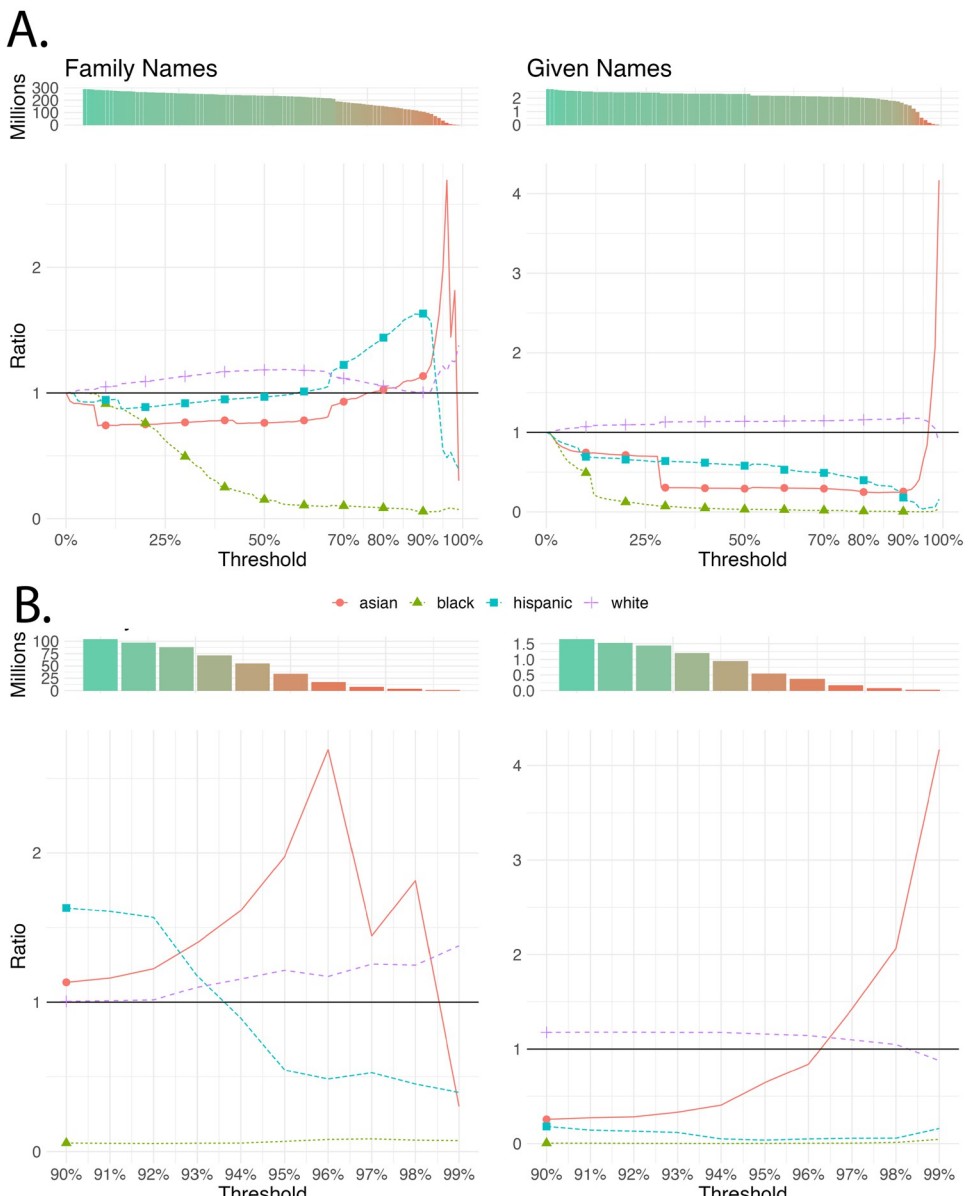

**Fig 3. Changes in groups share, and people retrieved, by threshold.** Census (Family names) and mortgage (Given names) datasets. The evolution of thresholds between 0 and 1 (A), and detail on thresholds between 0.9 and 1 (B).

coinciding with the Black Power movement [36], distinctively Black first names became increasingly popular, particularly among Black people living in racially segregated neighborhoods [37].

## The effect of thresholding

Fig 4 shows the effect of using a 90% threshold on the WoS dataset of unique authors. The first column (A) corresponds to each author counting fractionally towards each racial category in proportion to the probabilities of their name distribution, using family names from the census, i.e., this is the closest we can get to ground truths with the available information. The

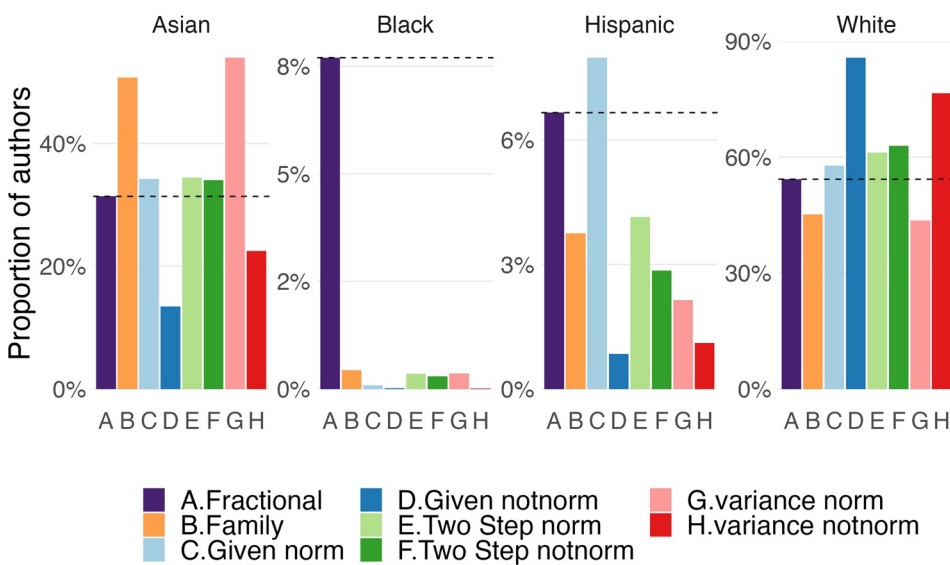

**Fig 4. Resulting distribution on different models with 90% threshold.** Fractional counting on family names for comparison.

remaining columns represent inference based on family (B) and given names (C-D) alone; the two-steps strategy, using both normalized (E) and unnormalized (F) given names, and the merged distributions of given and family names, with normalized (G) and not normalized (H) given names; always with a 90% threshold. All models severely under-represented the Black population of authors. Compared to the fractional baseline (A), all models except normalized given names (C) under-represent the Hispanic population. The unnormalized given name, either alone (D) or in the variance model (H), under-represents the Asian population. Finally, the White population is over-represented by all models except family names and the variance with normalized given names.

Fig 5 shows the seven different models' evolution over the threshold. First, the number of retrieved authors as the threshold increases; second, the ratio between the proportion a group represents given a model and a threshold, and the proportion using the fractional counting with family names. The dashed line represents the expected total cases per group using fractional counting, and the unbiased ratio of 1, respectively. A high threshold is expected to retrieve less cases than the expected total. For thresholds until 80%, this is not always the case for White authors. This means that for the two-step strategy, for a threshold below 80%, we would overestimate the total number of White authors. For Asian authors, given names have the worst retrieval, whereas Hispanic and especially Black authors are always underestimated. The retrieved authors fall sharply for all models after the 95% threshold.

As in Fig 4, we can compare for a given threshold the aggregate proportion of authors in each group, with respect to the expected ground truth. In this case, we can see that almost every model overestimates the proportion of White authors until the 90% or 95% thresholds, where Asian authors begin to be overestimated. Again, Hispanic and especially Black authors are heavily underestimated, with the single exception of the normalized given names, that overestimate Hispanic authors in the thresholds between 90% and 95%.

We conclude from this that a threshold-based approach, while intuitive and straightforward, should not be used for racial inference. Rather, analysis should be adapted to consider

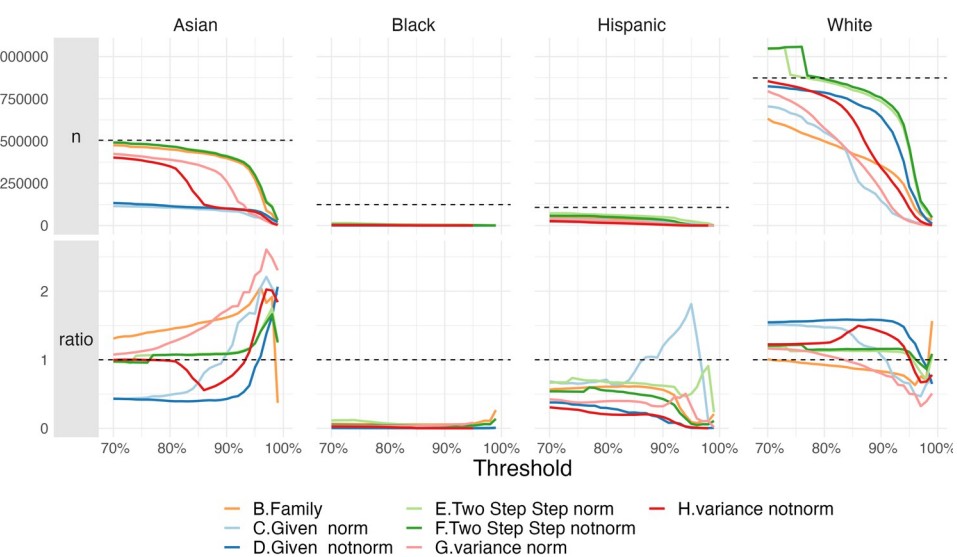

**Fig 5. Retrieval of authors by race using different inference models for varying thresholds.**

each author as a distribution over every racial category; in this way, even though an individual cannot be assigned into a category, aggregate results will be less biased.

## The effect of imputation

Another consideration is how to deal with unknown names. As mentioned in the Data section, the family names dataset provided by the Census Bureau covers 90% of the U.S. population. The remaining 10%, as well as author names not represented in the census, represents 774,381 articles, or 18.75% of the dataset, for which the family name of the first authors has an unknown distribution over racial categories.

An intuitive solution would be to impute missing names with a default distribution based on the racial composition of the entire census. Alternatively, the "All other names" category provided by the U.S. Census could be used. Table 3 shows the distribution among racial groups in the U.S. Census, the "All other names" category, and in WoS for first authors with family names included in the U.S. Census data. The Asian population is highly over-represented among WoS authors, whereas Hispanic and Black authors are highly under-represented, with respect to their proportion of the U.S. population. Imputing with the census-wide racial distribution or the special wildcard category is, therefore, equivalent to skewing the distribution towards Hispanic and Black authors and under-representing Asian authors. Since the ground truth is contingent to the specific dataset in use, a better imputation would instead be the mean of the population most representative of an individual. For example, in the case of a missing author name in the WoS, the racial distribution of that individual's discipline could be

**Table 3. Racial distribution in U.S. Census and WoS U.S. Authors with known family names.**

| Racial group | U.S. Census aggregate | U.S. Census "All other names" | U.S. WoS |
|---|---|---|---|
| Asian | 5.0% | 8.2% | 24.5% |
| Black | 12.4% | 8.8% | 7.2% |
| Hispanic | 16.5% | 14.1% | 5.4% |
| White | 66.1% | 68.8% | 59.4% |

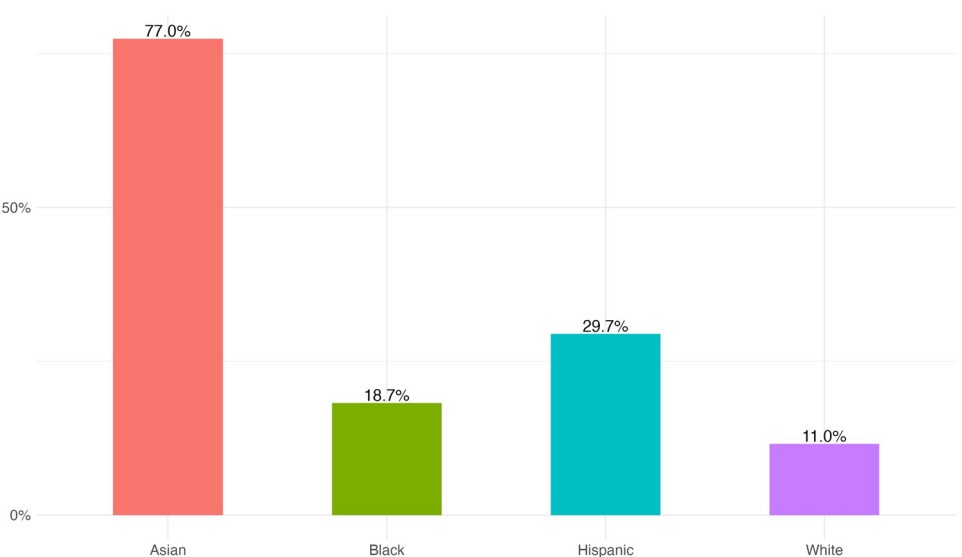

**Fig 6. Proportion of Temporary Visa Holders by racial group.**

imputed. Our recommendation is -in cases where imputation is needed- to first compute the aggregate distribution of racial categories with the dataset in which the inference is intended, and then use this aggregate distribution to impute in those family names missing from the census dataset. Statistically, this preserves the aggregate distribution on this dataset.

Nevertheless, this type of imputation can also introduce new biases. If the missing family names correlate with a specific racial group, then the known cases cannot be considered a random sample of the data, and their mean will be biased toward those groups that have fewer unknown names. Knowing which group has more unknown cases is in principle an impossible task. Nevertheless, it is possible to infer this, considering the citizenship status of authors. Authors that are temporary visa holders in US are more likely to have a family name that doesn't appear on the census. The Survey of Earned Doctorates provides information on doctorate recipients, by ethnicity, race, and citizenship status between 2010 and 2019 [38]. Fig 6 shows the average proportion of Temporary Visa Holders among Earned Doctorates from each racial group. This can be seen as a proxy of the distribution of authors by race and citizenship status. There is a large majority of Asian authors that are migrants, followed by a 30% of Hispanic authors, 19% of Black authors and 11% of White authors. Imputing by the mean of the known authors would also underestimate Asian authors, and partially too Hispanic authors, while overestimating White authors. Nevertheless, omitting the missing cases would have the same effect on the overall distribution, given that the imputation by the mean does not change the aggregate proportion of each group. There is no perfect solution for this, as the distribution shown on Fig 6 is only a proxy of the problem. Therefore, it is important to acknowledge this potential bias on the result, both if the imputation is used or if the missing cases are omitted.

## Conclusion

Race scholars [39] have advocated for a renewal of Bourdieu's [40] call for reflexivity in science of science [41]. We pursue this through empirical reflexivity: challenging the instrumentation used to collect and code data for large-scale race analysis. In this paper we manually validate and propose several approaches for name-based racial inference of U.S. authors. We

**Table 4. General recommendations for implementing a name-based inference of race for U.S. authors.**

| | Do's | Don'ts |
|---|---|---|
| *Given Names* | Use only family names from U.S. Census to avoid bias. | Do not use given names, except when the underlying distribution of your dataset matches that of mortgage data. |
| *Thresholding* | Consider each person in your data as a distribution and adapt your summary statistics. | Do not use a threshold for categorical classification of each person, as this under-represents Black population, due to the correlation between racial groups and name informativeness. |
| *Imputation* | If needed, calculate first the aggregated distribution on your dataset, and use this for imputation of missing cases. Acknowledge the potential bias of imputation. | Do not use the census aggregate distribution for imputation, except when your target population matches the U.S. population. |

demonstrated the behaviour of the different methods on simulated data, across the population, and on authors in the WoS database. We also illustrated the risks of underestimating highly minoritized groups (e.g., Black authors) in the data when using a threshold, and the overestimation of White authors introduced by given names when they are based on mortgage data. A similar result was identified by Cook [10], in her attempt to infer race of patent data based on the U.S. Census: she found that the approach "significantly underpredicted matches to black inventors and overpredicted matches to white inventors" and concludes that the name-based inference approach was not suitable for historical analyses.

From our analysis, we come away with three major lessons that are generally applicable to the use of name-based inference of race in the U.S., shown in Table 4.

Inferring race based on name is an imperfect, but often necessary approach to studying inequities and prejudice in bibliometric data (e.g., Freeman & Huang, 2014), and in other areas where self-reported race is not provided. However, the lessons shown here demonstrate that care must be taken when making such inferences in order to avoid bias in our datasets and studies.

It has been argued that science and technology serve as regressive factors in the economy, by reinforcing and exacerbating inequality [42]. As Bozeman [42] argued, "it is time to rethink the economic equation justifying government support for science not just in terms of why and how much, but also in terms of who." Studies of the scientific workforce that examine race are essential for identifying who is contributing to science and how those contributions change the portfolio of what is known. To do this at scale requires algorithmic approaches; however, using biased instruments to study bias only replicates the very inequities they hope to address.

In this study, we attempt to problematize the use of race from a methodological and variable operationalization perspective in the U.S. context. In particular, we acknowledge variability in naming conventions over time, and the difficulty of algorithmically distinguishing Black from White last names in the U.S. context. However, any extension of this work across country lines will necessarily require tailoring to meet the unique contextual needs of the country or region in question. Ultimately, scientometrics researchers utilizing race data are responsible for preserving the integrity of their inferences by situating their interpretations within the broader socio-historical context of the people, place, and publications under investigation. In this way, they can avoid preserving unequal systems of race stratification and instead contribute to the rigorous examination of race and science intersections toward a better understanding of the science of science as a discipline. Once again, we quote Zuberi [3]: "The racialization of data is an artifact of both the struggles to preserve and to destroy racial stratification."

## Limitations

The name-based racial inference proposed in this article avoids individual identification of authors and instead uses the distribution of probabilities associated with each name. This has

limitations: for the two U.S. Census groups that account for a small proportion of the population—American Indian and Alaska Native (AIAN) and Two or more races—the inference power of the method is weak, and can lead to spurious results on the aggregate level. To avoid misleading results, we exclude these groups from the analysis and re-normalize the distribution. This is an acknowledged limitation of this work and—to the best of our knowledge—an unavoidable effect of algorithms that seek to infer race based on names. An alternative methodology would be to survey authors to obtain their self-declared race data to investigate racial inequalities in scholarly publications. However, given that individuals' identities are also critically important to protect, the distributional approach proposed in this article presents the advantage that it cannot be used to identify authors' race on an individual basis.

There is a pressing need for large-scale analyses of racial bias in science. That said, algorithmic approaches which fail to account for all minoritized and marginalized groups are limited. Therefore, this study demonstrates the need for complementary sets of quantitative and qualitative studies focused on the racialized identities of groups that would otherwise be excluded from large-scale studies such as the one presented here.

## Author Contributions

**Conceptualization:** Diego Kozlowski, Vincent Larivière, Thema Monroe-White, Cassidy R. Sugimoto.

**Data curation:** Diego Kozlowski, Dakota S. Murray, Alexis Bell, Will Hulsey, Vincent Larivière.

**Formal analysis:** Diego Kozlowski.

**Funding acquisition:** Diego Kozlowski, Vincent Larivière.

**Investigation:** Diego Kozlowski, Vincent Larivière.

**Methodology:** Diego Kozlowski, Vincent Larivière, Thema Monroe-White, Cassidy R. Sugimoto.

**Project administration:** Vincent Larivière, Thema Monroe-White.

**Resources:** Vincent Larivière.

**Software:** Diego Kozlowski.

**Supervision:** Vincent Larivière, Thema Monroe-White, Cassidy R. Sugimoto.

**Validation:** Diego Kozlowski, Dakota S. Murray, Alexis Bell, Will Hulsey, Vincent Larivière.

**Visualization:** Diego Kozlowski, Dakota S. Murray.

**Writing – original draft:** Diego Kozlowski, Vincent Larivière, Thema Monroe-White, Cassidy R. Sugimoto.

**Writing – review & editing:** Diego Kozlowski, Vincent Larivière, Thema Monroe-White, Cassidy R. Sugimoto.

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
