## [Decision Letter · Decision Letter 0]

25 Nov 2021

PONE-D-21-32854Avoiding bias when inferring race using name-based approachesPLOS ONE

Dear Dr. Kozlowski,

Thank you for submitting your manuscript to PLOS ONE. After careful consideration, we feel that it has merit but does not fully meet PLOS ONE’s publication criteria as it currently stands. Therefore, we invite you to submit a revised version of the manuscript that addresses the points raised during the review process. Please submit your revised manuscript by Jan 09 2022 11:59PM. If you will need more time than this to complete your revisions, please reply to this message or contact the journal office at plosone@plos.org. Please include the following items when submitting your revised manuscript:A rebuttal letter that responds to each point raised by the academic editor and reviewer(s). You should upload this letter as a separate file labeled 'Response to Reviewers'.A marked-up copy of your manuscript that highlights changes made to the original version. You should upload this as a separate file labeled 'Revised Manuscript with Track Changes'.An unmarked version of your revised paper without tracked changes. You should upload this as a separate file labeled 'Manuscript'.

We look forward to receiving your revised manuscript.

Kind regards,

Lutz Bornmann

Academic Editor

PLOS ONE

Journal Requirements:

“VL acknowledges funding from the Canada Research Chairs program, https://www.chairs-chaires.gc.ca/,  (grant # 950-231768), DK acknowledges funding from the Luxembourg National Research Fund, https://www.fnr.lu/, under the PRIDE program (PRIDE17/12252781).”

Reviewers' comments:

Reviewer's Responses to Questions

**Comments to the Author**

1. Is the manuscript technically sound, and do the data support the conclusions?

Reviewer #1: Yes

Reviewer #2: Yes

2. Has the statistical analysis been performed appropriately and rigorously? 

Reviewer #1: Yes

Reviewer #2: Yes

3. Have the authors made all data underlying the findings in their manuscript fully available?

Reviewer #1: Yes

Reviewer #2: Yes

4. Is the manuscript presented in an intelligible fashion and written in standard English?

Reviewer #1: Yes

Reviewer #2: Yes

5. Review Comments to the Author

Reviewer #1: The manuscript "Avoiding bias when inferring race using name-based approaches" discusses and analyzes the problems of bibliometric analyses that include racial classifications on a conceptual and methodological level. The authors conclude with dos and don'ts for such analyses.

Overall, the manuscript is well written and should be of interest to the readers of PLoS One.

I wonder about the section regarding imputation of missing data. Maybe, the authors could provide advice on whether imputation should be done at all. It seems to be taken for granted that some imputation method should be used. Is imputation from a proper distribution better than no imputation at all? Authors without proper assignment could be just removed from the data set. What would be the advantages and disadvantages?

Reviewer #2: The paper investigates different approaches for inferring the perceived race of authors in the U.S. This can serve as an important methodological basis for large-scale empirical analyses on racial inequalities in the science system. The paper presents a compelling approach to this research question and valuable conclusions for inferring perceived race in bibliometric data. However, I recommend to address the following points before publication.

152-158: At this point, it was difficult for me to get an idea of what the simulated data is used for. A sentence after "First, to test the interaction between given and family names distributions, we simulate a dataset that covers most of the possible combinations" could help to clarify this (e.g. something like "This step is only used to determine how to combine given and family names for inferring race").

169-170: "Questions now include both racial and ethnic origin, placing "Hispanic" outside racial categories. The racial categories in both datasets include Hispanic as a category, ...". At first sight, this sounds contradictory ("Hispanic" is not in racial categories, and at the same time "Hispanic" is used as a category). I would suggest to clarify here that "Hispanic" is not a racial category in the original US Census data, but you use it as a racial category in your datasets. There are also no quotes around "Hispanic" in line 170, while this is usually the case in line 169.

186-187: This implies that only first authors are considered for your analyses. This restriction should be mentioned explicitly, and also why you chose to do so (and did not include other author positions).

193-222: This part seems to better fit in the "Methods" section than the "Data" section. Unless there are good reasons to keep it in the "Data" section, you may want move this part.

226: In the "Methods" section, it is unclear which particular approaches you finally use in your empirical analyses. In particular, which of the three weighting schemes did you finally use for your analyses? I think a concise list of the approaches you use would be helpful for the reader.

252: Use "n" instead of "c" in the summation notation for consistency with the formula in line 245 (or vice versa).

254: "for both given (family) names" looks like it is a measure for two given names or two family names. But as far as I understand it, this weight combines the given and the family name of one person. Should this be "for both given and family names"?

259: Which value did you finally use for exp?

264-272: Which color pattern should be observed in order to have a good approach? Have you tried exponent values > 2? If not, why?

290: ";" -> "."

304-307: A more detailed explanation of how the given name distribution has been normalized and how this affects the results would be helpful here.

Figure 3: How are the values on the y-axis (ratio) calculated, and how does this measure over-/underrepresentation? The frequencies of given/family names that are shown in the upper plots provide important information, but the distribution is difficult to inspect for thresholds > 90%. Can this be visualized in a form that better shows this part of the distribution (e.g. in an extra figure)?

399-416: It is a good point and convincingly shown by the results presented here that simply imputing based on the Census data should be avoided. But imputing based on the distributions in the bibliometric data for known names may also introduce biases. This would be the case if the probability for missing names correlates with the race category (a reason for this might be the development over time of both the probability to have full names in the database and the distribution across race categories). I would argue one has to be very cautious when imputing bibliometric data, because usually these data only provide a limited amount of metadata that can be used for imputation. Given the usually large number of cases in bibliometric data, it is probably not necessary to impute data in most cases. I would argue that it is more important and transparent to discuss possible biases for a particular research question introduced due to missing data than trying to impute the data. I think your results also provide a very good basis for such a discussion with regard to inferring race.

6. PLOS authors have the option to publish the peer review history of their article (what does this mean?). If published, this will include your full peer review and any attached files.

Reviewer #1: No

Reviewer #2: No

---

## [Author Response · Author response to Decision Letter 0]

29 Nov 2021

Journal Requirements:

Answer:

We have modified the format to adapt it to PLOS ONE style. 

“VL acknowledges funding from the Canada Research Chairs program, https://www.chairs-chaires.gc.ca/, (grant # 950-231768), DK acknowledges funding from the Luxembourg National Research Fund, https://www.fnr.lu/, under the PRIDE program (PRIDE17/12252781).”

Answer:

We have add the amendment in the cover letter.

Reviewer #1:

I wonder about the section regarding imputation of missing data. Maybe, the authors could provide advice on whether imputation should be done at all. It seems to be taken for granted that some imputation method should be used. Is imputation from a proper distribution better than no imputation at all? Authors without proper assignment could be just removed from the data set. What would be the advantages and disadvantages?

Answer:

We thank the reviewer for this valuable question. Indeed, this was taken for granted in the original submission, but constitutes a very important debate. Both the imputation and the omission of unknown cases can generate a bias if the distribution of unknown cases is not random. We addressed this issue in the (new) Fig 6, and this new paragraph:

Nevertheless, this type of imputation can also introduce new biases. If the missing family names correlate with a specific racial group, then the known cases cannot be considered a random sample of the data, and their mean will be biased toward those groups that have fewer unknown names. Knowing which group has more unknown cases is in principle an impossible task. Nevertheless, it is possible to infer this, considering the citizenship status of authors. Authors that are temporary visa holders in US are more likely to have a family name that doesn’t appear on the census. The Survey of Earned Doctorates provides information on doctorate recipients, by ethnicity, race, and citizenship status between 2010 and 2019 [38]. Fig 6 shows the average proportion of Temporary Visa Holders among Earned Doctorates from each racial group. This can be seen as a proxy of the distribution of authors by race and citizenship status. There is a large majority of Asian authors that are migrants, followed by a 30% of Hispanic authors, 19% of Black authors and 11% of White authors. Imputing by the mean of the known authors would also underestimate Asian authors, and partially too Hispanic authors, while overestimating White authors. Nevertheless, omitting the missing cases would have the same effect on the overall distribution, given that the imputation by the mean does not change the aggregate proportion of each group. There is no perfect solution for this, as the distribution shown on Fig 6 is only a proxy of the problem. Therefore, it is important to acknowledge this potential bias on the result, both if the imputation is used or if the missing cases are omitted.

Reviewer #2:

152-158: At this point, it was difficult for me to get an idea of what the simulated data is used for. A sentence after "First, to test the interaction between given and family names distributions, we simulate a dataset that covers most of the possible combinations" could help to clarify this (e.g. something like "This step is only used to determine how to combine given and family names for inferring race").

Answer:

We added a clarifying sentence (170-172 in the highlighted version)

169-170: "Questions now include both racial and ethnic origin, placing "Hispanic" outside racial categories. The racial categories in both datasets include Hispanic as a category, ...". At first sight, this sounds contradictory ("Hispanic" is not in racial categories, and at the same time "Hispanic" is used as a category). I would suggest to clarify here that "Hispanic" is not a racial category in the original US Census data, but you use it as a racial category in your datasets. There are also no quotes around "Hispanic" in line 170, while this is usually the case in line 169.

Answer:

 We thank the reviewer for the suggestion, the data sources used on this manuscript add ‘hispanic’ as a racial category. We added a clarifying sentence (184-188 in the highlighted version) 

186-187: This implies that only first authors are considered for your analyses. This restriction should be mentioned explicitly, and also why you chose to do so (and did not include other author positions).

Answer:

Indeed, we only used first authors to be sure they were US-based. We added a clarification in lines 201-205 of the highlighted version 

193-222: This part seems to better fit in the "Methods" section than the "Data" section. Unless there are good reasons to keep it in the "Data" section, you may want move this part.

Answer:

We moved this part to the Methods section. 

226: In the "Methods" section, it is unclear which particular approaches you finally use in your empirical analyses. In particular, which of the three weighting schemes did you finally use for your analyses? I think a concise list of the approaches you use would be helpful for the reader.

Answer:

 We added a list of the methods used on the experiments at the end of the Methods section (317-323 of the highlighted version)

252: Use "n" instead of "c" in the summation notation for consistency with the formula in line 245 (or vice versa).

Answer:

 We thank the reviewer for noticing this, we fixed the inconsistency

254: "for both given (family) names" looks like it is a measure for two given names or two family names. But as far as I understand it, this weight combines the given and the family name of one person. Should this be "for both given and family names"?

Answer:

We fix the sentence to make it more clear

259: Which value did you finally use for exp?

264-272: Which color pattern should be observed in order to have a good approach? Have you tried exponent values > 2? If not, why?

Answer:

 We thank the reviewer for this comment, we explored values of the exponent 1 and 2. We further clarify why in the sentence 282-286 of the highlighted version.

290: ";" -> "."

Answer:

 We fixed the typo

304-307: A more detailed explanation of how the given name distribution has been normalized and how this affects the results would be helpful here.

Answer:

 We agree with the reviewer that a more detailed explanation was needed. We add a clarification in lines 337-346 of the highlighted version

Figure 3: How are the values on the y-axis (ratio) calculated, and how does this measure over-/underrepresentation? The frequencies of given/family names that are shown in the upper plots provide important information, but the distribution is difficult to inspect for thresholds > 90%. Can this be visualized in a form that better shows this part of the distribution (e.g. in an extra figure)?

Answer:

 We add further explanation of this on lines 353-356 of the highlighted version. We have also divided Figure 3 into A and B, where A is the original figure, and B shows a detailed version for thresholds above 90%

399-416: It is a good point and convincingly shown by the results presented here that simply imputing based on the Census data should be avoided. But imputing based on the distributions in the bibliometric data for known names may also introduce biases. This would be the case if the probability for missing names correlates with the race category (a reason for this might be the development over time of both the probability to have full names in the database and the distribution across race categories). I would argue one has to be very cautious when imputing bibliometric data, because usually these data only provide a limited amount of metadata that can be used for imputation. Given the usually large number of cases in bibliometric data, it is probably not necessary to impute data in most cases. I would argue that it is more important and transparent to discuss possible biases for a particular research question introduced due to missing data than trying to impute the data. I think your results also provide a very good basis for such a discussion with regard to inferring race.

 Answer:

We thank the reviewer for raising this very important issue. As we mentioned for reviewer #1, we agree this was not properly address and it can be a potential source of bias. Both the imputation by the mean and the omission of missing cases can be introduced bias if the distribution of missing name is not random. We added an explanation for this in lines 461-478 of the highlighted version, and we also added Fig 6. That shows the potential non-randomness of missing cases given by the citizenship status of authors.

---

## [Decision Letter · Decision Letter 1]

8 Feb 2022

Avoiding bias when inferring race using name-based approaches

PONE-D-21-32854R1

Dear Dr. Kozlowski,

We’re pleased to inform you that your manuscript has been judged scientifically suitable for publication and will be formally accepted for publication once it meets all outstanding technical requirements.

Kind regards,

Lutz Bornmann

Academic Editor

PLOS ONE

Additional Editor Comments (optional):

Reviewers' comments:

Reviewer's Responses to Questions

**Comments to the Author**

1. If the authors have adequately addressed your comments raised in a previous round of review and you feel that this manuscript is now acceptable for publication, you may indicate that here to bypass the “Comments to the Author” section, enter your conflict of interest statement in the “Confidential to Editor” section, and submit your "Accept" recommendation.

Reviewer #1: All comments have been addressed

Reviewer #2: All comments have been addressed

2. Is the manuscript technically sound, and do the data support the conclusions?

Reviewer #1: Yes

Reviewer #2: Yes

3. Has the statistical analysis been performed appropriately and rigorously? 

Reviewer #1: Yes

Reviewer #2: Yes

4. Have the authors made all data underlying the findings in their manuscript fully available?

Reviewer #1: Yes

Reviewer #2: Yes

5. Is the manuscript presented in an intelligible fashion and written in standard English?

Reviewer #1: Yes

Reviewer #2: Yes

6. Review Comments to the Author

Reviewer #1: (No Response)

Reviewer #2: (No Response)

7. PLOS authors have the option to publish the peer review history of their article (what does this mean?). If published, this will include your full peer review and any attached files.

Reviewer #1: No

Reviewer #2: No

---

## [Editor Report · Acceptance letter]

21 Feb 2022

PONE-D-21-32854R1 

Avoiding bias when inferring race using name-based approaches 

Dear Dr. Kozlowski:

I'm pleased to inform you that your manuscript has been deemed suitable for publication in PLOS ONE. Congratulations! Your manuscript is now with our production department. 

Kind regards, 

on behalf of

Dr. Lutz Bornmann 

Academic Editor

PLOS ONE